# An Electrolyte Life Indicator for Plasma Electrolytic Polishing Optimization

Facheng Su [1], Hsiharng Yang [1,2,*], Wenchieh Wu [3] and Yukai Chen [3]

1   Graduate Institute of Precision Engineering, National Chung Hsing University, Taichung City 40227, Taiwan
2   Innovation and Development Center of Sustainable Agriculture (IDCSA), National Chung Hsing University, Taichung City 40227, Taiwan
3   Intelligent Technology Development Section, Metal Industries Research and Development Centre, Kaohsiung City 82151, Taiwan
*   Correspondence: hsiharng@nchu.edu.tw

**Abstract:** This work shows that electrolyte current-density as an indicator can assist in the optimized timing of the addition of the electrolyte to plasma electrolytic polishing (PEP) to keep it active and in operation. In this experiment, 2 wt% ammonium sulfate was used as an electrolyte to polish 1 cm × 1 cm stainless steel SUS304. The hot-bath heating method was successfully used to heat it from 60 to 90 °C, followed by suction filtration. The cathode was fixed at the beaker edge in the electrolyte and the input voltage was 340 volts. Once the gas-phase layer formed stably around the workpiece, the plasma went through the electrolyte to polish the workpiece surface. Then, the anode was slowly immersed into the electrolyte and the current-density measured. It was found that based on the current-density–temperature curve, for the timing of the addition of the electrolyte, the current-density difference could be used to decide whether it needed to be supplemented or not. When the temperature was from 75 to 80 °C and 85 to 90 °C, it was found that the 2 wt% ammonium sulfate solution should be supplemented. The result showed that the electrolyte life indicator, using the current-density, is a feasible method of practical technology for PEP.

**Keywords:** plasma electrolytic polishing; electrolyte life indicators; metal polishing





## 1. Introduction

Plasma electrolytic polishing (PEP) is a combination of reaction processes that removes the surface of a metallic part via plasma-physical and electrochemical reactions [1]. Depending on the polished materials and the electrolyte, the voltage is usually set at a high level between 200 and 400 V in order to generate a plasma discharge. The polishing process and results are affected by many parameters, such as the electrolyte type, electrical source, pretreatment, cathode, workpiece, etc. Among those parameters, the treatment time, electrolyte, electrical source and voltage are the three main critical factors that affect surface roughness [1,2]. A high electrolyte temperature is also critical to gas layer formation. The continuous gas layer surrounding the sample does not appear in a low-temperature electrolyte. High temperature in an electrolyte plays a key role in the PEP system, formatting a steady and continuous gas-phase layer surrounding the surface sample [3].

Figure 1 shows the current–voltage characteristics of an anode in the polishing process, which is divided into four sections. Firstly, the A–B section can be represented by Ohm's law and Faraday's law, and it shows a positive correlation between the current and the voltage. When voltages rise at point B, vapor and gas are regularly formed into a film around workpiece but do not entirely cover it. Secondly, the B–C section can be described as unstable plasma electrolytic polishing. The gas-phase layer of the workpiece is rendered stable until the voltages increase to point C. Thirdly, the C–D section can be expressed as the stable PEP process. Lastly, the D–E area of the current becomes stabilized [4].

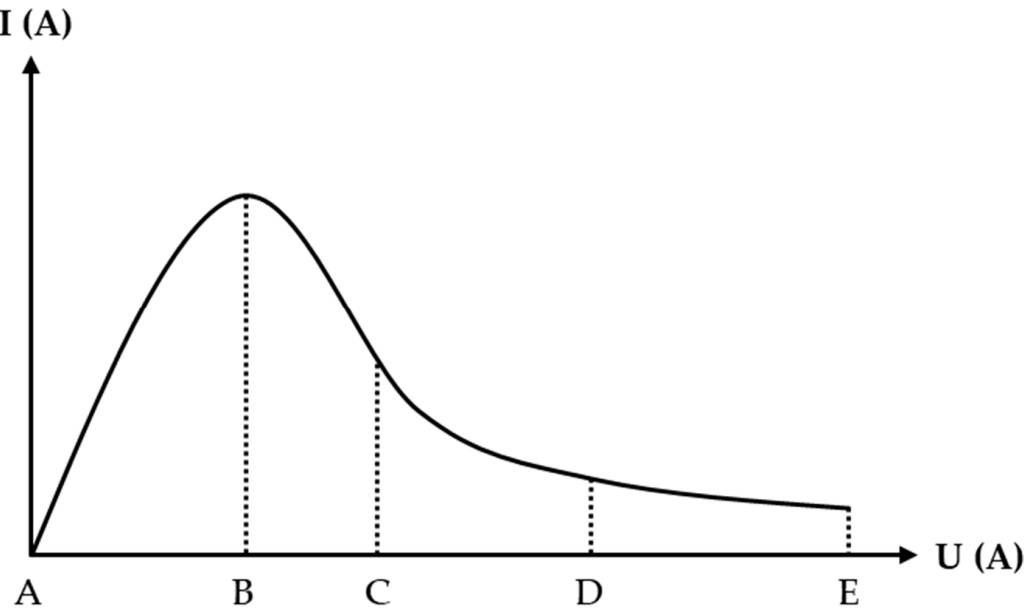

**Figure 1.** Current–voltage characteristics of the PEP anode process [5].

In the past, traditional polishing was extensively applied to medical instruments [6], and electronic devices and in aerospace. The purpose of traditional polishing is acquiring a low-roughness and high-gloss surface. Traditional polishing mainly includes mechanical polishing, chemical polishing and electrochemical polishing. Mechanical polishing involves making the surface smooth through plastic deformation; the disadvantages of this method are that it needs more workers to operate it, and it cannot be promised that each workpiece surface will be of the same quality. Chemical polishing can remove the workpiece surface through a chemical reaction. Electrochemical polishing is similar to chemical polishing in that the workpiece is immersed in a chemical solution, but it is processed under a current. The metal workpiece is is an anode, while the electric tool is a cathode. Applying a direct current (DC), the workpiece anode is dissolved into metallic ions and removed atom by atom [7]. However, a high concentration of a strong acid or a strong base is used as an electrolyte. During the reaction, harmful gases and liquids will cause damage to workers and their environment [8]. Thus, it will produce waste liquid after the reaction.

Unlike the strong acid and caustic alkali solutions used in chemical and electrochemical polishing, the electrolyte used in the plasma electrolytic polishing process is a low-concentration aqueous salt solution. The processing parameters used in the PEP process, such as electrolyte composition, temperature and high voltage applied to the metal alloys, result in the polishing effect of brought about by surface-material removal. The process is environmentally friendly and follows the safety requirements.

Plasma electrolytic polishing applications have already been successfully used in steel [9–13], aluminum [10,14], copper [15], titanium and their alloys [5]. However, during the PEP process, polishing ability will be decreased because the electrolyte is evaporated at high temperatures and participates in electrochemical action. Thus, polishing ability will be affected by factors such as conductivity, the generation of metal impurities during the polishing process, electrolyte concentrations and pH value changes. Therefore, its status must be monitored and maintained or replaced in the process. The objective of this study was to access the electrolyte life indicator by measuring the electrolyte, and to find the index that can represent the life of the electrolyte by establishing a determination curve of temperature vs. current-density. After acquiring the electrolyte life indicator, the operator can clearly know the timing of the addition of the electrolyte and keep the workpiece quality as high as possible during polishing.

## 2. Materials and Methods

### 2.1. Plasma Electrolytic Polishing

A summary of the PEP process is shown in Figure 2. The polished workpiece is an anode and connected to the plus pole of the DC energy source, to which a high voltage is applied; meanwhile, the cathode is connected to the minus pole of the DC energy source. At the start of the plasma electrolytic polishing process, the electrolyte is electrolyzed. The electrolytic reaction happens near the anode, along with oxygen evolution and metal oxidation. The Me is the metal workpiece element, which is expressed in the following equations:

$$2H_2O + 4e^- \rightarrow O_{2\uparrow} + 4H^+$$

$$Me\text{-}ne^- \rightarrow Me^{n+};$$

where Me represents the metal workpiece element.

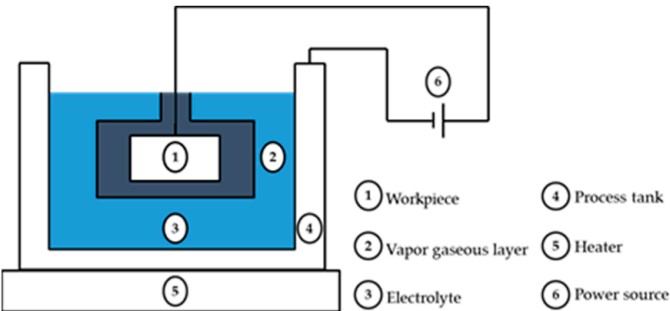

**Figure 2.** Schematic of plasma electrolytic polishing.

During polishing, a stable gas-phase layer is formed between the workpiece surface and the electrolyte, meaning that the electric circuit is disconnected. Meanwhile, the resistance is increased between the workpiece and the electrolyte, which forms a high voltage. Discharge bombardment happens in the gas-phase layer. Then, plasma is produced in the discharge channel, causing the metallic and gas-phase layers to produce strong plasma-physical and electrochemical reactions [1]; this lets a chemical reaction be produced and an electrical discharge removes the surface of the polished workpiece [6,16]. When the discharge removal velocity is bigger than the chemical-reaction-producing rate, surface peaks are removed and a smoother surface is achieved [14,17,18].

### 2.2. Experimental Setup of Plasma Electrolytic Polishing

Before polishing, the test piece was immersed into acetone; then, ultrasonication was used to dissolve and remove surface grease and other organic impurities. The parameters of the experiment are shown in Table 1. The electrolyte used was 2 wt% ammonium sulfate (Shen Chiu Enterprise Co., Taichung, Taiwan) and 340 volts were applied to polish the 1 cm × 1 cm stainless steel SUS304 workpiece (Regional R&D Service Department, Taichung, Taiwan). The chemical compositions are shown in Table 2.

**Table 1.** Experimental parameters.

| Parameters | Value |
|---|---|
| Voltage | 340 V |
| Electrolyte | Ammonium sulfate |
| Electrolyte concentration (wt%) | 2% |
| Electrolyte temperature | From 60 to 90 °C |
| Workpiece | Stainless steel (SUS 304) |
| Diving depth | 30 mm |

**Table 2.** Chemical composition of SUS304 [17,19–21].

| Element | Percentage (%) |
| --- | --- |
| Carbon | $\leq 0.08$ |
| Silicon | $\leq 1.00$ |
| Manganese | $\leq 2.00$ |
| Phosphorus | $\leq 0.045$ |
| Sulfur | $\leq 0.03$ |
| Chromium | 18.00~20.00 |
| Nickel | 8.00~10.50 |

Figure 3a shows the workpiece, on which the anode is located in the central part of the beaker, while the cathode is attached to the border of beaker. Figure 3b is a closer look at the anode and cathode. Then, the hot-water heating method was used to heat the electrolyte from 60 °C to 90 °C for the purpose of maintaining a stable system and undergoing less energy loss. In order to form a stable gas-phase layer between the workpiece surface and the electrolyte, a voltage of 340 volts was applied to the workpiece, which was regulated by a DC power supply (ADC 4000, Preen, Taipei, Taiwan), as shown in Figure 3c. Finally, the current-density was measured from 60 °C to 90 °C at 5 °C intervals.

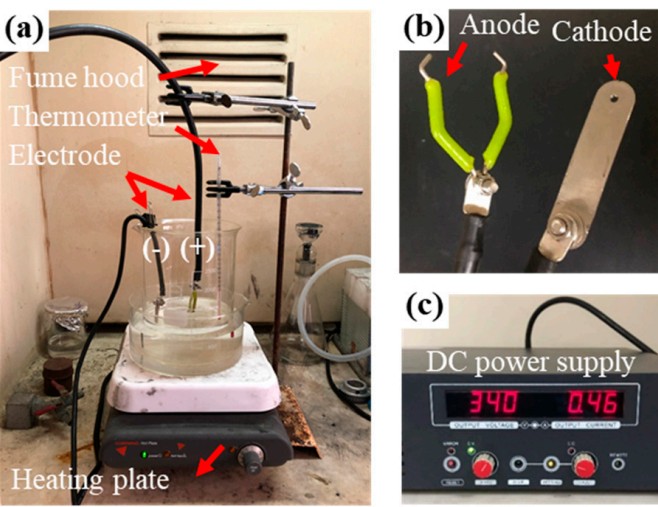

**Figure 3.** Experimental device for plasma electrolytic polishing: (**a**) PEP system; (**b**) zoomed in observation of electrode; (**c**) DC power supply.

Current-density is defined as the amount of current flowing through an electrode per unit area which can be represented by the following equation:

$$i = \left(\frac{I}{A}\right) = \left(\frac{Ke \cdot \Delta U}{g}\right) \tag{1}$$

where the $I$ is current ($A$), $A$ is the cross-sectional area which the current passes through ($m^2$), $Ke$ is the conductivity of the electrolyte, $\Delta U$ is the potential difference between the two electrodes and $g$ is the electrode gap.

## 3. Results and Discussion

### 3.1. Plasma Electrolytic Polishing

### 3.1.1. Color Changes of Polished Liquid

Figure 4 shows the process of plasma electrolytic polishing. It can be clearly seen that the color of the polishing liquid was changed from transparent to dark as time went on. Figure 4a shows the hot-bath heating method that was used to maintain a system stable; however, there was a 10 °C temperature difference between the inside and the outside of the beaker. As shown in Figure 4b, the solution was mixed with the metallic part of the workpiece and the electrolyte. Nevertheless, it was better to analyze the residual element using energy-dispersive X-ray spectroscopy. The color became darker because of the reaction combined with the plasma-physical and electrochemical reactions. In Figure 4c, electrolyte precipitation of the precipitate occurred. From Figure 4d to Figure 4f, suction filtration of the electrolyte was applied after electrolyte precipitation. With the suction filtration, the precipitate was removed.

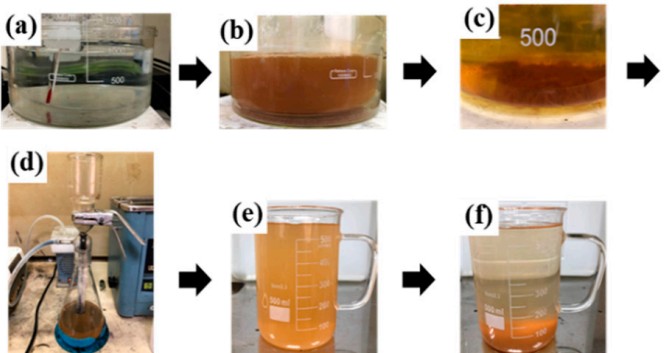

**Figure 4.** Process of plasma polishing process: (**a**) unpolished, the color is transparent; (**b**) during polishing; (**c**) after polishing, precipitation for 5 min; (**d**) suction filtration of the polishing liquid; (**e**) collection of polishing liquid after suction filtration; (**f**) precipitation of filtrated polish.

As shown in Figure 5, the use of the suction filtration method can effectively filter out solid sediment from the polishing liquid. After polishing, residues were generated from the reaction between the workpiece and the electrolyte. The goal of using the suction filtration method was to keep the best polishing condition of the 2 wt% ammonium sulfate solution. The fewer the residues produced from the workpiece and polishing solution, the higher the quality can become.

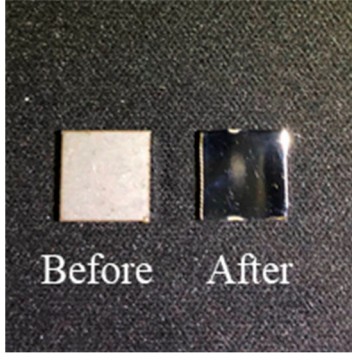

**Figure 5.** Contrast of workpiece before and after polishing.

### 3.1.2. Contrast of Workpiece before and after Polishing

Figure 6 shows the contrast as a result of polishing the 1 cm × 1 cm stainless steel SUS304, proving that the workpiece was well polished. Thereby, the surface gloss is apparently improved. In this experiment we found that it is important to slowly drop the workpiece in order to let the applied voltage and current become stable. On the other hand, if the workpiece is dropped directly, it is unable to reach the predetermined voltage; moreover, the gas-phase layer cannot form on the surface of the workpiece.

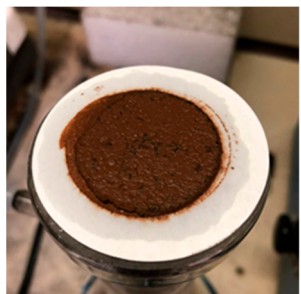

**Figure 6.** Schematic of residues on filter paper after suction filtration.

Table 3 compares the plasma electrolytic polishing process parameters to previous work. Compared to previous work, this work used the hot-bath heating method to maintain system stability, and the use of the suction filtration method enabled the reuse of the electrolyte.

**Table 3.** Comparison of the plasma electrolytic polishing process parameters.

| Metallic Material | Electrolyte (wt%) | Voltage (V) | Reuse of Electrolyte | Temperature (°C) | Hot-Bath Heating Method and Suction Filtration Method | Reference |
|---|---|---|---|---|---|---|
| Stainless steel SUS304 | 0.4% $(NH)_2SO_4$ | 280 | None | 79–81 | No | Zong [22] |
| Stainless steel SUS304 | 3% $(NH)_2SO_4$ | 280 | None | 17 | No | Wang [21] |
| Stainless steel SUS304 | 2% $(NH)_2SO_4$ | 340,340 | None | 60–90 | Yes | This work |

### 3.2. Current-Density–Temperature Curve

Figure 7 presents the density–temperature curve resulting from the use of 2 wt% ammonium sulfate. Table 4 shows the measured data from the experiment. The difference in the current-density drops down dramatically from 0.729 to 0.49 A/cm$^2$ at the temperature of 60 to 65 °C. As known from the current-density–temperature characteristic curve in Figure 1, this stage has unstable polishing conditions, which mean the gas-phase layer does not entirely cover the whole workpiece. From a temperature of 65 to 70 °C, the variation in the current-density decreases to 0.032 A/cm$^2$ as this stage is stable in the PEP process. However, the current-density difference increases to 0.075 A/cm$^2$ from 75 to 80 °C which could indicate that the PEP process returns from a stable stage to an unstable stage. Thus, the electrolyte should be supplemented. Between 85 and 90 °C, the current-density difference is 0.057 A/cm$^2$, which is slightly higher than the current-density difference of 0.034 A/cm$^2$ at 80 to 85 °C, so it is necessary to add electrolyte at this interval. In addition to finding out the electrolyte life index, the average consumption rate of the polishing liquid is 7.08 mL/min when the temperature is maintained at 90 °C.

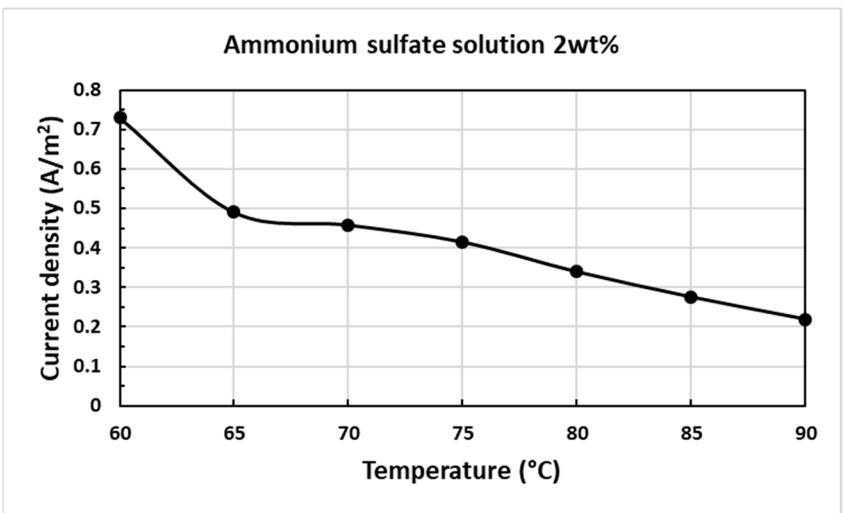

**Figure 7.** Supplementary curve of current-density–temperature curve resulting from the use of 2 wt% ammonium sulfate.

**Table 4.** Measurement of current-density from 60 to 90 °C.

| Temperature (°C) | Current-Density (A/cm$^2$) |
| --- | --- |
| 60 | 0.729 |
| 65 | 0.49 |
| 70 | 0.458 |
| 75 | 0.415 |
| 80 | 0.34 |
| 85 | 0.276 |
| 90 | 0.219 |

## 4. Conclusions

This paper provided a practical method using a 2 wt% ammonium sulfate current-density–temperature curve and improved two PEP processes. First, the hot-bath heating method can increase the temperatures' uniformity and stability in the polishing liquid; further, the hot-bath heating method in the electrolyte maintenance mechanism can be expanded to future production lines. Second, the use of the suction filtration method can rapidly separate the sediment from the polishing liquid. Therefore, the suction filtration method in electrolyte cleaning and the maintenance mechanism can be extended to future production lines. Furthermore, it is crucial to conduct a study of the variable electrolyte life indicator after the application of PEP in the future.

**Author Contributions:** Conceptualization, Y.C.; methodology, F.S. and H.Y.; software, Y.C.; validation, H.Y., W.W. and Y.C.; formal analysis, F.S.; investigation, F.S.; resources, H.Y. and W.W.; data curation, F.S. and Y.C.; writing—original draft preparation, F.S.; writing—review and editing, H.Y.; visualization, H.Y.; supervision, Y.C.; project administration, H.Y. and W.W.; funding acquisition, H.Y. and W.W. All authors have read and agreed to the published version of the manuscript.

**Funding:** This research was funded by the Ministry of Science and Technology of Taiwan (MOST-110-2221-E-005-052-MY3) and also partially supported by the Ministry of Education, Taiwan, R.O.C. under the Higher Education SPROUT Project.

**Institutional Review Board Statement:** Not applicable.

**Informed Consent Statement:** Not applicable.

**Data Availability Statement:** Data sharing is not applicable to this article.

**Acknowledgments:** The joint research project between the Metal Industries Research and Development Centre and the National Chung Hsing University should be acknowledged.

**Conflicts of Interest:** The authors declare no conflict of interest.

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
