# Peer review of "An Electrolyte Life Indicator for Plasma Electrolytic Polishing Optimization"

_applsci, doi:10.3390/app12178594_

Round 1

Reviewer 1 Report

General comments: The communication does not bring significant scientific added value. The study is similar to practice work of students in laboratory.

Line 64-76: The paragraph related to pbjective of the study is too long and contains conclusions. The last part (from Line 66-76) should be relocated at the end of the study in conclusion section if the authors can prove all these assumptions. If these statements represent results from other studies then please add references.

Line 84-88: Please rephrase. You used the expression The Me is the metal workpiece element twice

Line 98-104 and Figure 2: This is not part of authors study, so it cannot be introduced in Materials and Methods section. Please move this paragraph in Introduction section.

Figure 3 caption: This is not a schematic of PeP. The picture represents an experimental device. Please make the corrections.

Line 123-128: I think you should delete the sub-section title 2.3. The lines from 123 to 128 should be part of sub-section 2.2

Line 132-135: Where are the the discussions? What happen there?

Line 144: In table 1 are presented your work compared with other studies. What are the differences and how these differences influence the quality of the products/process .... or which are the advantages?

Line 145-149: I cannot understand your assumption. What represents the curve from Figure 8? Is it the perfect evolution of the current density or a representation of the values from Table3? In case of perfect model, could you add other curves obtained from your experimental procedure? 

Author Response

All comments are answered and revised in the manuscript. Thanks for your patient to review our manuscript.

Reviewer 2 Report

This study is related to plasma electrolytic polishing (PEP) of stainless steel SUS304. The work is very good and subject matter of the journal. However, before publishing authors should address the comments given below.

1) Please write everywhere PEP instead of PeP

2) Line 32, please write voltage is usually set ---- instead of voltage usually set-----

3) Line 33, please write -----results are affected by ---- instead of ------ results are effect by ----

4) Line 37, please write ------are affected--- instead of ------are effected----.

5) Line 41, please write ---is mainly--- instead of ----was mainly---

6) Lines 43,44, please reframe the sentence ---higher require of people----quality. It is confusing.

7) Lines 44,45, please reframe the sentence ----- is take advantage of ------material. It is confusing.

8) Line 49, please write is instead of was.

9) Line 51, please write workers.

10) Line 55, 56, what is metal material?

11) Line 61, ----polishing ability will be decreased such as conductivity--- is confusing. Are the factors to decrease the ability? Pleasereframe the sentence. What is the meaning of contains polished ----? Please check the sentence.

12) Lines 67-72, there is no verb, please reframe the sentence.

13) How did authors obtain chemical composition in Table 2? Please write about the source or experimental procedure.

14) Line 126, please write cross-sectional instead of crossing-sectional.

15) Line 144, it will be Table 4. Authors should discuss about their work and previous works. What type stainless steel were used in previous works?

16) Line 152,155, It will be Table 5, not Table 3.

17) Authors should discuss about the advantages of present work compared to previous works as they have written in lines 64-76.

18) Authors should write the role temperature in PEP in introduction section.

19) English is very bad, it has to be improved a lot before publishing. 

Author Response

(The authors gave the same response as above.)

Round 2

Reviewer 1 Report

Thank you for updating your work.